# Bivariate Causal Discovery via Conditional Divergence

**Bao Duong**    DUONGNG@DEAKIN.EDU.AU  and  **Thin Nguyen**    THIN.NGUYEN@DEAKIN.EDU.AU
*Applied Artificial Intelligence Institute ($A^2I^2$), Deakin University, Australia*

**Editors:** Bernhard Schölkopf, Caroline Uhler and Kun Zhang

## Abstract

Telling apart cause and effect is a fundamental problem across many science disciplines. However, the randomized controlled trial, which is the golden-standard solution for this, is not always physically feasible or ethical. Therefore, we can only rely on passively observational data in such cases, making the problem highly challenging. Inspired by the observation that the conditional distribution of effect given cause, also known as the causal mechanism, is typically invariant in shape, we aim to capture the mechanism through estimating the stability of the conditional distribution. In particular, based on the inverse of stability – the divergence – we propose Conditional Divergence based Causal Inference (**CDCI**), a novel algorithm for detecting causal direction in purely observational data. By doing this, we can relax multiple strict assumptions commonly adopted in the causal discovery literature, including functional form and noise model. The proposed approach is generic and applicable to arbitrary measures of distribution divergence. The effectiveness of our method is demonstrated on a variety of both synthetic and real data sets, which compares favorably with existing state-of-the-art methods.

**Keywords:** Causal discovery, causal direction, conditional divergence

## 1. Introduction

Distinguishing cause from effect is both a fascinating and highly complex problem across many scientific disciplines. Knowing the underlying causal mechanism of nature allows us to precisely predict the outcomes of any imaginable interventions. Ideally, by actively intervening on the variables of interest then analyzing feedback data, the randomized controlled trials (RCT) enable us to discriminate cause from effect in the most reliable way. However, the imperfect reality disallows us to have such convenient tool available for all interesting problems. Performing RCT experiments in real life is majorly impossible due to the limitations of resource, expenses, or ethical issues. Therefore in such cases, passively observational data is the only clue we have to disentangle the hidden mechanism of reality. In contrary to RCTs, predicting causal relationships using solely observational data is exceedingly challenging, and sometimes impossible. Recent years have seen countless breakthroughs in the field of machine learning (ML), with the spectacular ability to capture non-linear and highly complex relationships even with unstructured data (e.g., image, text, audio, etc.). However, current ML systems merely learn correlations from observational data, without understanding the real nature.

For those reasons, the scientific community has been continuously trying to solve the problem and pushing the boundary of human knowledge about causality mechanisms, which has resulted in a variety of well-developed algorithms. In this work, we exclusively focus on the two-variable instance of the problem. That is, given a finite sample of two univariate variables $X$ and $Y$, determine whether $X$ causes $Y$ ($X \rightarrow Y$), $Y$ causes $X$ ($Y \rightarrow X$), or there is no causal relationships between them. This problem setting has also received enormous attentions from researchers in the last

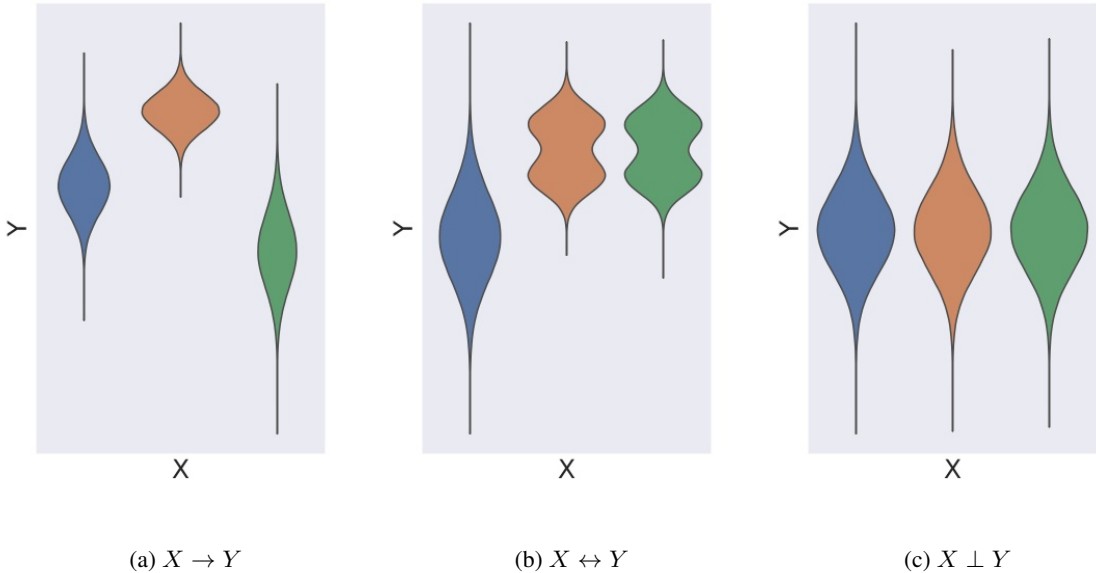

(a) $X \to Y$         (b) $X \leftrightarrow Y$         (c) $X \perp Y$

Figure 1: Simple causal direction orientation rule based on the similarity of conditional distributions: (a) Similar shapes, different locations and scales $\Rightarrow X$ causes $Y$, (b) Different shapes, locations or scales $\Rightarrow X$ and $Y$ have a common confounder, (c) Similar shapes, locations and scales $\Rightarrow X$ and $Y$ are independent.

decades. The majority of these methods base on the formalism of functional causal models (FCM) (Pearl, 2009), where the effect is assumed to be generated from the cause via a functional mechanism. Methods following this formalism aims to exploit the inherent asymmetry between cause and effect, and employ an array of different assumptions ranging from realistic to highly restrictive. Linear non-Gaussian acyclic models (LiNGAM) (Shimizu et al., 2006), additive noise models (ANM) (Hoyer et al., 2008), and post-nonlinear model (Zhang and Hyvärinen, 2009) address the problem by assuming a functional form on the causal mechanism, as well as a noise model. In information geometric causal inference (IGCI), Daniušis et al. (2010) pay attention to a more limited scenario, where the causal mechanism is assumed to be invertible with little or no noise allowed. Other exciting frameworks have also been applied against this challenge, such as minimum description length (MDL) principle (Marx and Vreeken, 2017; Mian et al., 2021), generative neural networks (GNN) (Goudet et al., 2018), and meta learning (Ton et al., 2021).

**Present work.** In detecting potential binary causal relations, we follow the assumption that the causal mechanism should be stable for the true causal direction (Janzing and Schölkopf, 2010). In other words, if $X \to Y$ is the true causal relation, for example, then the conditional distribution of the effect given the cause $P(Y|X)$ is independent with the marginal distribution of the cause $P(X)$. This is in accordance with the postulate that the shape of the conditional distribution $P(Y|X = x)$ is invariant for different values of $x$ (Fonollosa, 2019). Particularly, in this paper, we propose to capture the stability of the conditional distribution, by measuring its inverse – the divergence of conditional distributions, with a novel algorithm called conditional divergence based causal infer-

ence (**CDCI**).[1] The gist of this idea is that the more the conditional distributions of the assumed effect $P(Y|X)$ diverge, conditioned on the assumed cause $(X)$, the less likely $X \rightarrow Y$ is the true relation between the two variables. Figure 1 illustrates the motivation of our approach, where a measure of conditional divergence can be jointly combined with an independence test to derive a useful and understandable rule for orienting causal relations.

Several variants of **CDCI**, different in the divergence measures, are implemented, including Chi-squared distance (Pearson, 1900), Hellinger distance (Diaconis and Zabell, 1982), Kullback-Leibler divergence (Kullback and Leibler, 1951), Kolmogorov metric (Kolmogorov, 1933), and total variation distance (Gibbs and Su, 2002). We compare the performance of our methods against state-of-the-art baselines. The experimental results demonstrate that our **CDCI** variants are able to consistently outperform traditional causal direction detectors on a variety of simulated and real datasets. For instance, on a mixed simulated and real dataset, while the best baseline gains an AUC of 0.742, our proposed **CDCI** methods achieve a much higher AUC, ranging from 0.865 to 0.884.

**Contributions.** The key contributions of our study can be summarized as follows:

1. We propose an approach to estimate the average divergence of normalized conditional distributions. This approach is generic, which smoothly accommodates several well-found measures of distribution divergence.

2. We provide a set of conditional divergence based predictors of causal direction.

3. We demonstrate the effectiveness of the proposed conditional divergence based methods against state-of-the-arts in predicting causal directions on both simulated and real data sets.

The paper is structured as follows: In section 2 we present the key concepts of causal discovery and highlights the state-of-the-arts in the field. Next, section 3 explains the proposed **CDCI** method that solves the bivariate causal discovery problem based on the stability of conditional distributions. Then, section 4 discusses fundamental identifiability analyses of **CDCI**. Subsequently, section 5 presents the experimental settings and main results on the effectiveness of our method in predicting causal orientation. Finally, in section 6 we review the contributions of our study and provide directions for future developments.

## 2. Background

Here we highlight general approaches that have been tried to address the causal direction prediction problem or similar issues.

### 2.1. Preliminaries

Following the FCM formalism (Pearl, 2009), we assume that the causal relationship is non-bidirectional and the effect is generated as a function of the cause and a noise:

$$Y := f(X, \epsilon)$$

where $\epsilon$ is the background noise associated with $Y$. In many formalizations, $X$ and $\epsilon$ are assumed to be independent. However, in this work we allow for the dependence between them in acknowledgment for cases that the effect is generated by both the cause and a confounder, for example:

---

1. Relevant source code and data sets can be found at https://github.com/baosws/CDCI

$$X := g\left(Z, \epsilon_X\right)$$
$$Y := f\left(X, Z, \epsilon_Y\right)$$

where $Z$ is a confounder, i.e., the common cause between $X$ and $Y$. That is, our method aims to detect the true causal direction despite the presence of confounders.

Then, the goal of differentiating cause and effect problem is to classify an empirical bivariate joint distribution $P(X, Y)$ into one of three main categories: $X \to Y$ ($X$ causes $Y$), $Y \to X$ ($Y$ causes $X$), and 'non-causal' relationship, such as $X \leftrightarrow Y$ (neither $X$ or $Y$ causes the other but both are influenced by another factor) and $X \perp Y$ ($X$ and $Y$ are independent). Following the Cause-Effect Pairs Challenge 2013 (Guyon and Statnikov, 2019) and many closely related works (Fonollosa, 2019; Goudet et al., 2018, 2019; Mooij et al., 2016; Ton et al., 2021), three classes $X \to Y$, $Y \to X$, and 'non-causal' are numerically encoded into $1$, $-1$ and $0$, respectively. The prediction is allowed to be any real number, which can be interpreted as the ranking score, where positive values stand for $X \to Y$, negative values suggest $Y \to X$, and zero indicates no causal relationships.

## 2.2. Related works

### 2.2.1. MULTIVARIATE METHODS

From a wider perspective, the causal relation prediction task is not generally restricted to only two variables. There have been great successes in finding causal relationships among a causal network of multiple variables (Spirtes et al., 2000; Chickering, 2002; Tsamardinos et al., 2006; Ogarrio et al., 2016; Bühlmann et al., 2014; Mian et al., 2021; Dhir and Lee, 2020). One of the traditional approaches is constraint-based methods (Spirtes et al., 2000), which exploit the conditional independence relationships among variables to recover the causal structure. However, the conditional independence tests are not applicable to the bivariate case because they require at least one more variable to condition on, apart from the two we have, and they cannot always orient all edges in a causal network.

Alternatively, score-based methods (Chickering, 2002) search for the causal graph with highest fitting score, which can be computationally intensive and inaccurate. Hybrid methods (Tsamardinos et al., 2006; Ogarrio et al., 2016), which combine constraint-based and score-based approaches, have been shown to be more accurate in many situations than the original approaches. In general, to orient causal directions, the majority of multivariate methods rely greatly on the global structure of the causal network, which cannot be exploited in the case of only two variables.

### 2.2.2. ASYMMETRY BETWEEN CAUSE AND EFFECT

In the bivariate setting, methods inspired by the notion of asymmetry between cause and effect are the most widely recognized. A foundational approach is proposed by Dodge and Rousson (2000, 2001), who exploit the correlation coefficient in linear models to develop asymmetric indicators (e.g., skewness ratio) that can tell apart the explanatory and the response variables.

Another early and successful attempt to realize this idea is based on the additive noise assumption (Shimizu et al., 2006; Hoyer et al., 2008; Shimizu et al., 2011; Zhang and Hyvärinen, 2009; Mooij et al., 2016). These methods assume that $Y$ is generated as a function of $X$ plus an independent noise: $Y := f(X) + \epsilon, X \perp \epsilon$. Many approaches have followed this direction and partially

exploited the linear (Shimizu et al., 2006, 2011), non-linear (Hoyer et al., 2008), and post-nonlinear relationships (Zhang and Hyvärinen, 2009). Recently, (Blöbaum et al., 2019) show that if both variables are equivalently scaled and the functional mechanism is nearly deterministic, then the regression error is smaller in the true causal model, so the causal direction can be inferred by comparing the residuals after fitting a function (e.g., polynomial, logistic function, or neural network) on both possible causal directions.

In addition, under the "equal variance" assumption, where the noise terms have a common variance, identifiability can be ensured in several models by exploiting the order of variables induced by their variances (Peters and Bühlmann, 2013; Chen et al., 2019; Park and Kim, 2020).

Overall, the aforementioned methods typically impose very strong assumptions on the functional mechanism family and the properties of the noise variables, which are usually not readily available in practice, thus making them very limited for a vast variety of applications.

### 2.2.3. LEARNING-BASED APPROACHES

A recent class of methods attempts to adapt learning algorithms to classify causal directions. In randomized causation coefficient (RCC) (Lopez-Paz et al., 2015) and neural causation coefficient (NCC) (Lopez-Paz et al., 2017), causality is posed as a classification problem, where the input bivariate distribution is featurized using either kernel mean embeddings or neural networks, which is subsequently used as the input for a binary classifier.

Apart from that, non-linear SEM estimation using non-stationarity (NonSENS) (Monti et al., 2020) and causal mosaic (Wu and Fukumizu, 2020) base their approaches on the non-linear independent component analysis methods, implemented by neural networks. Causal generative neural networks (CGNN) (Goudet et al., 2018) adopts generative neural networks to model the causal relationships for both directions, then compare them to decide the causal direction. Meta-CGNN (Ton et al., 2021) extends this idea with meta learning to allow for the employment of similar data sets to predict causal direction of unseen data.

### 2.2.4. INDEPENDENCE BETWEEN CAUSE AND MECHANISM

Most similar to our method is a branch of works that rely on the postulate called 'independence between cause and mechanism', which states that for the true causal direction $X \rightarrow Y$, the conditional distribution of effect given cause $P(Y|X)$ is independent of the marginal distribution $P(X)$, but not for the opposite direction. Information geometric causal inference (IGCI) (Daniušis et al., 2010) assumes an invertible causal mechanism, with low to no noise. The authors suppose that the effect is less uniform than the cause, which leads them to define the measure of independence between cause and mechanism using the entropies of $X$ and $Y$.

Another line of works postulates that if $X \rightarrow Y$ is the true causal direction then the total Kolmogorov complexities (Grünwald and Vitányi, 2008) of the marginal distribution $P(X)$ and the conditional distribution $P(Y|X)$ should be less than the other way around: $K(P(X)) + K(P(Y|X)) < K(P(Y)) + K(P(X|Y))$ (Janzing and Schölkopf, 2010; Stegle et al., 2010; Marx and Vreeken, 2017), rendering Occam's razor philosophy: the simpler explanation is more likely the true one. Since the Kolmogorov is not computable, they upper bound it by either the minimum message length (MML) principle (Stegle et al., 2010) or minimum description length (MDL) principle (Marx and Vreeken, 2017; Mian et al., 2021).

In causal inference via kernel deviance measures (Mitrovic et al., 2018), the authors develop a fully non-parametric method based on the independence of Kolmogorov complexity of the mechanism in the causal direction and the cause, in which they use the framework of reproducing kernel Hilbert spaces (RKHS) to measure the variability of Kolmogorov complexities of the mechanism. An alternative assumption is that if $X \to Y$ is the underlying causal relation, then the shapes of conditional distributions of effect given cause $P(Y|X = x)$ should be similar for different values of $x$ (Fonollosa, 2019), where the standard deviation of conditionals distributions $P(Y|X)$ is used as the the direction score.

## 3. Conditional Divergence based Causal Inference (CDCI)

### 3.1. Conditional Divergence

**Assumption.** We assume that if $X \to Y$ is the true causal direction then the cause, i.e., $X$, is independent with the geometrical shape of the conditional distribution of the effect $Y$ given the cause $X$, i.e., $P(Y|X)$. We emphasize that this assumption is a generalization of several existing assumptions, including:

- Independence between cause and noise: for instance, the generating process $Y := X + \epsilon$ with $\epsilon \sim \mathcal{U}(0, X)$ satisfies our assumption even though $X \not\perp \epsilon$.

- Additive Noise Models ($Y := f(X) + \epsilon, X \perp \epsilon$) and Multiplicative Noise Models ($Y := \epsilon \times f(X), X \perp \epsilon$): If a bivariate joint distribution $P(X, Y)$ admits one of these models, then the conditional distributions of $Y$ given $X$ only differ in either location or scale, respectively.

- Functional form: we do not impose any major restriction on the causal mechanism family. That being said, **CDCI** is fully non-parametric and involves a richer family of models.

To realize this idea, we propose to measure the invariance of conditional distributions shapes. We practically achieve this by estimating the divergence of conditional distributions. For this purpose, we define a conditional distribution divergence measure called *Normalized Conditional Divergence (NCD)*, which estimates the average distance of conditional distributions from a central one. Generally, *NCD* of $Y$ given $X$ can be mathematically formulated as:

$$NCD_D(Y|X) := \mathbb{E}_{x \sim P(X)} \left[ D \left( P(\widehat{Y}|X = x), P(\widehat{Y}) \right) \right] \tag{1}$$

where $D$ is any valid probability distance measure (e.g., Kullback-Leibler divergence), and $\widehat{Y}$ is a random variable generated for the $k$-th sample $\left( x^{(k)}, y^{(k)} \right)$ as follows:

$$\widehat{y}^{(k)} := \frac{y^{(k)} - \mathbb{E}\left[ Y|X = x^{(k)} \right]}{\sqrt{\mathrm{Var}\left[ Y|X = x^{(k)} \right]}}$$

This is essentially the standardization of the conditional distribution, which helps eliminate both location and scale variations, allowing *NCD* to be influenced by the distribution shape only. In our implementation, any continuous variable is discretized before computation as follows: first, outliers with deviations greater than a specified $max\_dev$ standard deviations are grouped, then the resulting range is divided into $4 \times max\_dev + 1$ equally spaced bins.

Since *NCD* adopts the notion of probability distance, the higher values indicate the higher dissimilarity in the shape of conditional distributions, whereas the minimum value of zero is reached when all conditional distributions share an identical shape. Regarding the choices for the probability distance $D$, in subsection 3.3 we showcase some options for this as variants of our method and demonstrate that high performance is achievable regardless of the choice.

## 3.2. Conditional Divergence based Causal Score

---

**Algorithm 1** The **C**onditional **D**ivergence based **C**ausal **I**nference **(CDCI)** algorithm for causal direction prediction based on conditional distribution divergence.

---

**Input:** An i.i.d. bivariate joint distribution $P(X, Y)$, and a probability distance $D$.
**Output:** The causal score $C_{X \to Y}$ and direction.
1. Compute the conditional divergences for both directions using the probability distance $D$, as in Equation (1):

$$C_{X|Y} := NCD(X|Y)$$
$$C_{Y|X} := NCD(Y|X)$$

2. Compute the causal score:

$$C_{X \to Y} := C_{X|Y} - C_{Y|X}$$

3. Output the causal score $C_{X \to Y}$ and

$$\text{direction} := \begin{cases} X \to Y & \text{if } C_{X \to Y} > 0 \\ Y \to X & \text{if } C_{X \to Y} < 0 \\ \text{Non-causal} & \text{if } C_{X \to Y} = 0 \end{cases}$$

---

To determine the causal direction, only the conditional distribution divergence on one direction is not enough (i.e., $Y$ given $X$). More particularly, it is usually subjective and not trivial to tell whether an arbitrary conditional divergence is too small or too large. For those reasons, we furthermore devise a conditional divergence based causal score as the direct indicator for the causal direction. The causal score for direction $X \to Y$ is given by:

$$C_{X \to Y} := NCD(X|Y) - NCD(Y|X) \tag{2}$$

The conditional divergence based causal score is a powerful causal direction indicator and clearly interpretable: a positive value means that the conditional distributions of $Y$ given $X$ are more similar and hence $X \to Y$ is more likely to be the true orientation, while a negative value suggests the opposite, and the value of zero is a sign of two variables being independent. The intuition is as follows: if $X$ is a true cause of $Y$, then the conditional distributions $P(Y|X)$ diverges less

Table 1: Probability Distances and Corresponding **CDCI** Variants.

| Probability distance | Symmetric? | Distance range | CDCI variant | Score range |
|---|---|---|---|---|
| Chi-squared distance | No | $[0; +\infty)$ | CCS | $(-\infty; +\infty)$ |
| Hellinger distance | Yes | $[0; \sqrt{2}]$ | CHD | $[-\sqrt{2}; \sqrt{2}]$ |
| Kullback-Leibler divergence | No | $[0; +\infty)$ | CKL | $(-\infty; +\infty)$ |
| Kolmogorov metric | Yes | $[0; 1]$ | CKM | $[-1; 1]$ |
| Total Variation distance | Yes | $[0; 1]$ | CTV | $[-1; 1]$ |

in shape, entailing $NCD(Y|X)$ is lower than $NCD(X|Y)$, making $NCD(X|Y) - NCD(Y|X)$ positive. The prediction causal direction can then be obtained with a decision threshold of approximately zero. To sum up, Algorithm 1 illustrates the big picture of our framework.

### 3.3. Variants of CDCI

Here we briefly describe five variants of **CDCI** where each one adopts a different probability metric commonly used in statistics and machine learning (see Gibbs and Su (2002) for a wider range of metrics and their interconnections). Each considered distance measure takes two probability density functions $P$ and $Q$ as input and returns their distance:

- **Variant 1: CCS** for the Chi-squared distance (Pearson, 1900):

$$D_{CS}(P, Q) := \sum_x \frac{(P(x) - Q(x))^2}{Q(x)}$$

- **Variant 2: CHD** for the Hellinger distance (Diaconis and Zabell, 1982):

$$D_{HD}(P, Q) := \sqrt{\sum \left( \sqrt{P(x)} - \sqrt{Q(x)} \right)^2}$$

- **Variant 3: CKL** for the Kullback-Leibler divergence (Kullback and Leibler, 1951):

$$D_{KL}(P, Q) := \sum_x P(x) \log \frac{P(x)}{Q(x)}$$

- **Variant 4: CKM** for the Kolmogorov metric (Kolmogorov, 1933):

$$D_{KM}(P, Q) := \max_x |P(X \leq x) - Q(X \leq x)|$$

- **Variant 5: CTV** for the Total Variation distance (Gibbs and Su, 2002):

$$D_{TV}(P, Q) := \frac{1}{2} \sum_x |P(x) - Q(x)|$$

For each of those variants, the placeholder $D$ in equation (1) is substituted with its corresponding probability metric. Table 1 presents some properties of the **CDCI** variants and their associated distance measures.

## 4. Identifiability

Using **CDCI**, the identifiability of the causal direction is ensured if exactly one of $P(Y|X)$ and $P(X|Y)$ is invariant in shape, which can be determined via the closed-form expressions of the conditional distributions. For example, if $P(Y|X)$ and $P(X|Y)$ are Gaussian or Uniform, then regardless of parameters, the causal direction is unidentifiable. In what follows, we investigate the identifiability of a family of causal models where the shape invariance is already met in the true direction.

**Proposition 1.** *Any causal model $Y := f(X) + \epsilon \times g(X)$ with $X \perp \epsilon$, $g \not\equiv 0$, and $P(\epsilon)$ is symmetrical has $P(Y|X)$ invariant in shape.*

**Proof.** The conditional normalization of $Y$ is $\widehat{Y}|_{X=x} = \frac{Y - \mathbb{E}[Y|X=x]}{\sqrt{\mathrm{Var}[Y|X=x]}} = \pm \frac{\epsilon - \mathbb{E}[\epsilon]}{\sqrt{\mathrm{Var}[\epsilon]}}$, which follows a symmetrical distribution independent of $X$. Furthermore, if $g(x) > 0 \ \forall x \in supp(X)$ or $g(x) < 0 \ \forall x \in supp(X)$ then $P(Y|X)$ is shape-invariant regardless of the asymmetry of $P(\epsilon)$.

While Proposition 1 characterizes a causal model family satisfying the shape invariance criteria for the true direction, it does not guarantee the opposite for the false direction. To see this, consider $g(X) \equiv 1$, $X$ and $\epsilon$ are Gaussian, and $f(X)$ is linear, then the model reduces to the well-known linear Gaussian additive noise model, which is unidentifiable since both $P(Y|X)$ and $P(X|Y)$ are Gaussian with the same bell-curve shape. Essentially, if the shape of $P(X|Y)$ is dependent on $Y$ (i.e., not invariant) then we can distinguish the cause from the effect.

Consequently, given the specification of a generative model following the form characterized by Proposition 1, we can determine its identifiability by analytically examining the conditional normalization $\widehat{X}|_{Y=y} = \frac{X - \mathbb{E}[X|Y=y]}{\sqrt{\mathrm{Var}[X|Y=y]}}$. If the closed-form simplification of $\widehat{X}|_{Y=y}$ involves $y$ then we can conclude that the shape of $P(X|Y)$ is not invariant, hence identifiability is ensured, and vice versa. Following directly from this procedure, over simple configurations such that $f, g \in \{\text{Constant}, \text{Linear}\}$ and $P(X), P(\epsilon) \in \{\text{Uniform}, \text{Gaussian}\}$, only two cases $(f, g, P(X), P(\epsilon)) \in \{(\text{Linear}, \text{Constant}, \text{Gaussian}, \text{Gaussian}), (\text{Linear}, \text{Constant}, \text{Uniform}, \text{Uniform})\}$ make the causal direction not recoverable due to $P(X|Y)$ being Gaussian and Uniform, respectively. This suggests that in general, with more complex settings, $P(X|Y)$ is likely to be shape-dependent on $Y$ as we desire for the false direction, so **CDCI** can correctly predict the causal direction in most situations.

## 5. Experiments

To demonstrate the effectiveness of our proposed **CDCI** approach, we conduct intensive experiments on a broad variety of both synthetic and real data sets. In this section, we first describe the experiment settings, including data sets, baseline methods, and evaluation criterion. Then, we provide results on the performance of our method in comparison with state-of-the-arts and the stability of our method under different settings. Finally, we conclude the section with some discussions.

### 5.1. Synthetic Data

For synthetic data sets, we employ the following five different data sets, which are also used in closely similar experiment settings from (Goudet et al., 2018; Ton et al., 2021):

**CE-Cha** (Guyon et al., 2019): 300 pairs of both continuous variables from the Cause-Effect Pairs Challenge 2013 (Guyon and Statnikov, 2019), with labels restricted to either $X \rightarrow Y$ or $Y \rightarrow X$.

**CE-Multi** (Goudet et al., 2018): The CE-Multi data set includes 300 generated pairs of continuous variables, with post-additive noise ($Y := f(X) + \epsilon, X \perp \epsilon$), pre-additive noise ($Y := f(X + \epsilon), X \perp \epsilon$), post-multiplicative noise ($Y := f(X) \times \epsilon, X \perp \epsilon$), and pre-multiplicative noise ($Y := f(X \times \epsilon), X \perp \epsilon$) models.

**CE-Net** (Goudet et al., 2018): CE-Net is comprised of 300 numerical pairs generated using neural networks with multiple choices for the noise distribution of the cause, including Exponential, Gamma, log-normal, and Laplace distributions.

**CE-Gauss** (Mooij et al., 2016): This data set consists of 300 numerical pairs artificially generated as follows: $X := f_X(\epsilon_X)$ and $Y := f_Y(X, \epsilon_Y)$ where $\epsilon_X$ and $\epsilon_Y$ are randomly drawn from Gaussian mixture distributions, whereas $f_X$ and $f_Y$ are randomly generated Gaussian processes.

**CE-All:** a concatenation of all four data sets above, creating a total of 1200 data pairs of continuous variables.

The target set for all those data sets is $\{X \to Y, Y \to X\}$ with roughly balance quantities. For each pair we have 1500 samples. All four data sets are publicly available and can be freely downloaded (Goudet et al., 2018).[2]

### 5.2. Real Data

In addition to synthetic data sets, we furthermore evaluate our methods and baselines on five real data sets from the Dialogue for Reverse Engineering Assessments and Methods challenge[3], ninth edition (referred to as DREAM4) in 2009 (Marbach et al., 2009, 2010; Prill et al., 2010), where the ground-truth causal networks are available. In the challenge, participants are asked to infer the topology of regulatory networks given gene expression data. Each data set includes a ground truth transcriptional regulatory network of *Escherichia coli* and *Saccharomyces cerevisiae*, as well as observations of gene expression measurements. For causal orientation task, we extract all directed edges and several pairs with no connection from the networks to create data sets containing all three causal classes $X \to Y, Y \to X$, and 'non-causal' (which includes both sub-classes $X \leftarrow Z \to Y$ and $X \perp Y$), with equal proportions for each class. Details for the data sets used are described as follows:

- **D4-S1** is from the sub-challenge one of the contest, containing 36 variable pairs with 105 samples for each pair.

- **D4-S2A**, **D4-S2B**, and **D4-S2C** are from the second sub-challenge, with 528, 747, and 579 pairs of variables, respectively, and each pair contains 210 samples.

- **D4-All** is a concatenation of the above four data sets, creating a total of 1890 pairs.

All the data sets are publicly accessible and can be freely downloaded.[4]

---

2. Data is available at https://dataverse.harvard.edu/dataset.xhtml?persistentId=doi:10.7910/DVN/3757KX

3. https://dreamchallenges.org

4. Data is available at https://www.synapse.org/#!Synapse:syn3049712/wiki/74628

Figure 2: Causal Direction Prediction Performance on Synthetic Datasets. The performance metric is AUC (higher is better). We compare five variants of our proposed **CDCI** approach namely CCS, CHD, CKL, CKM, and CTV (in red) with four popular baselines (in blue): ANM (Mooij et al., 2016), CDS (Fonollosa, 2019), IGCI (Daniušis et al., 2010), and RECI (Blöbaum et al., 2019).

### 5.3. Baseline Methods and Evaluation Metric

Similarly to several related studies (Goudet et al., 2018; Wu and Fukumizu, 2020; Ton et al., 2021), we compare the predictive power of our methods, including CCS, CHD, CKL, CKM, and CTV variants, against four competing methods which are briefly described below:

**Additive noise model (ANM)** (Mooij et al., 2016) with Gaussian Process mechanism and HSIC test for independence between cause and residuals.

**Conditional distribution similarity (CDS)** (Fonollosa, 2019) measures the variability of conditional distributions via standard deviation. The variability of conditional distributions for both $(X, Y)$ and $(Y, X)$ are compared to decide the causal direction.

**Information geometric causal inference (IGCI)** (Daniušis et al., 2010) compares the entropies of two variables and sets the one whose lower entropy as the cause..

**Regression error based causal inference (RECI)** (Blöbaum et al., 2019) fits a monomial of degree three to the normalized variables then compare the goodness of fit for both directions, in terms of mean squared error.

The implements of the Causal Discovery Toolbox (Kalainathan et al., 2020) for these methods are used with default parameters.

For performance metric, we use the bi-directional area under the receiver operating characteristic curve (AUC, the higher the better), which is the average of two AUC scores. The first AUC score measures the performance in classifying whether a pair $(X, Y)$ belongs to $X \rightarrow Y$ class or not, while the second AUC score accounts for the class $Y \rightarrow X$.

Regarding parameters for our methods, based on sample size, we use $max\_dev = 3$ and $max\_dev = 2$ for the discretization step in synthetic and real data, respectively.

### 5.4. Results

#### 5.4.1. SYNTHETIC DATA

Figure 2 shows how our methods consistently outperform all other methods across all synthetic data settings (for detailed figures, see Table 3 in the Appendix). Our best methods always achieve higher AUC compared with competing baselines. Even the one with worst performance of our methods

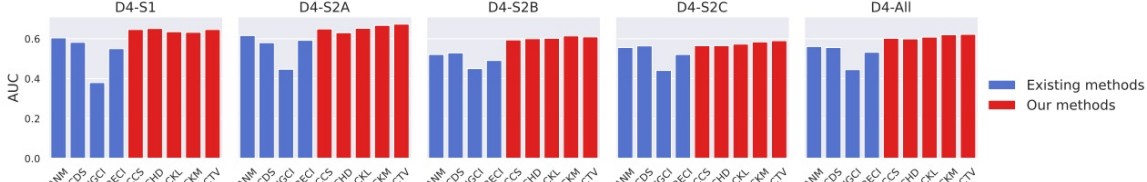

Figure 3: Causal Direction Prediction Performance on Real Datasets. The performance metric is AUC (higher is better). We compare five variants of our proposed **CDCI** approach namely CCS, CHD, CKL, CKM, and CTV (in red) with four popular baselines (in blue): ANM (Mooij et al., 2016), CDS (Fonollosa, 2019), IGCI (Daniušis et al., 2010), and RECI (Blöbaum et al., 2019).

Table 2: Causal Direction Prediction Performance w.r.t. Different Discretizations. The performance metric is AUC. We consider five probability distances as variants of **CDCI**, and five values for the discretization parameter $max\_dev \in \{1, 2, 3, 4, 5\}$ on synthetic (CE-All) and real (D4-All) data sets. *Overall* values are *mean $\pm$ standard deviation*.

| $max\_dev$ | CCS | CHD | CKL | CKM | CTV | Overall |
|---|---|---|---|---|---|---|
| | | | Synthetic data | | | |
| 1 | 0.719 | 0.733 | 0.719 | 0.752 | 0.727 | $0.73 \pm 0.01$ |
| 2 | 0.857 | 0.873 | 0.860 | 0.851 | 0.872 | $0.86 \pm 0.01$ |
| 3 | 0.866 | 0.884 | 0.867 | 0.865 | 0.881 | $0.87 \pm 0.01$ |
| 4 | 0.867 | 0.881 | 0.865 | 0.863 | 0.878 | $0.87 \pm 0.01$ |
| 5 | 0.860 | 0.874 | 0.858 | 0.863 | 0.875 | $0.87 \pm 0.01$ |
| Overall | $0.83 \pm 0.06$ | $0.85 \pm 0.07$ | $0.83 \pm 0.06$ | $0.84 \pm 0.05$ | $0.85 \pm 0.07$ | $0.84 \pm 0.06$ |
| | | | Real data | | | |
| 1 | 0.587 | 0.582 | 0.588 | 0.605 | 0.593 | $0.59 \pm 0.01$ |
| 2 | 0.602 | 0.599 | 0.608 | 0.620 | 0.622 | $0.61 \pm 0.01$ |
| 3 | 0.607 | 0.607 | 0.611 | 0.621 | 0.623 | $0.61 \pm 0.01$ |
| 4 | 0.657 | 0.648 | 0.659 | 0.659 | 0.667 | $0.66 \pm 0.01$ |
| 5 | 0.637 | 0.627 | 0.639 | 0.648 | 0.648 | $0.64 \pm 0.01$ |
| Overall | $0.62 \pm 0.03$ | $0.61 \pm 0.03$ | $0.62 \pm 0.03$ | $0.63 \pm 0.02$ | $0.63 \pm 0.03$ | $0.62 \pm 0.03$ |

dominates or is as good as the top competing methods in three out of five cases (CE-Net, CE-Gauss, and CE-All). The most significant difference can be seen in the highly representative data set CE-All, where CKM, with AUC of 0.865, surpasses the best baseline score of 0.742 (RECI), despite being the worst performer among variants of **CDCI** in this case. Additionally, we are able to maintain an adequately high lower bound of 0.7 in all synthetic data sets, while the performance of baseline methods can be as low as 0.16.

### 5.4.2. REAL DATA

Figure 3 demonstrates that even for real data, our methods are still capable of stably outperforming existing baselines (for detailed figures, see Table 3 in the Appendix). The results show that variants of **CDCI** can provide up to 16% of AUC improvement in comparison with the best baseline methods (in D4-S2B). Not only that, the worst performers of our variants also have higher scores than the best

state-of-the-arts in all five cases, with the most significant difference can be observed in D4-S2B, where the AUC scores of our methods are at least $12\%$ higher than that of competing methods.

### 5.4.3. INFLUENCE OF HYPERPARAMETERS

Our proposed method is influenced by two hyper-parameters, namely the probability metric - as indicated by **CDCI** variants, and the discretization parameter $max\_dev$. If $max\_dev$ is too small, it can lead to only a few discretized values (5 values for $max\_dev = 1$), which may be insufficient to capture the conditional distributions accurately. Meanwhile, if $max\_dev$ is set too large then our methods may suffer from outliers. To investigate the effect of hyper-parameters on the performance of our methods, in Table 2 we compare the AUC scores of **CDCI** variants across different values of $max\_dev$ in both synthetic and real data. The empirical results show that there is a visible degrade in performance of all **CDCI** variants for $max\_dev = 1$ and synthetic data, which may be in accordance with our expectation. However, on the remaining scenarios, the AUC score exhibits no significant variation across hyperparameters, where the standard deviations over all settings for synthetic and real data are only $1\%$ and $4\%$ of the average scores, respectively.

However, $max\_dev = 4$ should be opted for since it results in the highest average scores (over **CDCI** variants) in both synthetic and real data, with the numbers being $0.87 \pm 0.01$ and $0.66 \pm 0.01$, respectively. As for the best **CDCI** variant, CTV has the highest average AUC scores (over different values of $max\_dev$) in both synthetic and real data sets, which are $0.85 \pm 0.07$ and $0.63 \pm 0.03$, respectively, thus it can be preferred over other variants in practice.

### 5.5. Discussions

Theoretically, having a richer function class should lead to more unidentifiable cases, where both directions simultaneously satisfy the assumption so the causal direction cannot be decided. Yet, first theoretical analyses and empirical results have suggested that these cases are rare and does not significantly affect the effectiveness of our methods.

The advantages of our method are three-fold. First, it is generic and adaptable with alternative measures of divergence. The experimental results have confirmed that **CDCI** is robust regardless of the choice of divergence measures. Second, it does not require training, thus it is fast and always ready to be used for unseen data. Finally, it is insensitive to hyper-parameters and does not rely on randomness as in random sampling methods, hence allowing for reproducibility.

## 6. Conclusion

To realize the idea that a causal mechanism is stable, we base on the inverse of stability – divergence – to build divergence based causal direction predictors namely **CDCI** (Conditional Divergence based Causal Inference). We test **CDCI** with five variants, different in the measure of divergence, for the task of causal direction prediction. We benchmark the performance of our proposed methods against state-of-the-art approaches on a variety of synthetic and real data sets. We find that all the variants of **CDCI** perform really well on the data sets and are able to consistently outperform existing state-of-the-art solutions.

Future work should involve theoretical aspects of **CDCI**, wherein the identifiability results can be more concretely investigated, as well as extending **CDCI** to multivariate causal discovery, of which the bivariate causal relation inference studied in this work is a special case.

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

## Appendix A. Detailed Experimental Results

| Data set | Baselines | | | | CDCI | | | | |
|---|---|---|---|---|---|---|---|---|---|
| | ANM | CDS | IGCI | RECI | CCS | CHD | CKL | CKM | CTV |
| Synthetic data | | | | | | | | | |
| CE-Net | 0.865 | *0.884* | 0.574 | 0.660 | **0.897** | **0.905** | **0.893** | **0.943** | **0.919** |
| CE-Multi | 0.263 | 0.413 | 0.778 | *0.948* | **0.960** | **0.976** | **0.955** | 0.906 | **0.958** |
| CE-Gauss | 0.884 | *0.905* | 0.160 | 0.710 | 0.905 | **0.914** | **0.910** | **0.916** | **0.918** |
| CE-Cha | *0.705* | 0.695 | 0.556 | 0.590 | 0.693 | **0.720** | 0.698 | 0.697 | **0.722** |
| CE-All | 0.653 | 0.713 | 0.512 | *0.742* | **0.866** | **0.884** | **0.867** | **0.865** | **0.881** |
| Real data | | | | | | | | | |
| D4-S1 | *0.604* | 0.582 | 0.380 | 0.550 | **0.646** | **0.651** | **0.635** | **0.632** | **0.646** |
| D4-S2A | *0.616* | 0.580 | 0.447 | 0.592 | **0.649** | **0.630** | **0.652** | **0.667** | **0.673** |
| D4-S2B | 0.521 | *0.529* | 0.450 | 0.491 | **0.594** | **0.600** | **0.602** | **0.614** | **0.609** |
| D4-S2C | 0.556 | *0.564* | 0.441 | 0.521 | **0.565** | **0.565** | **0.573** | **0.584** | **0.590** |
| D4-All | *0.561* | 0.556 | 0.445 | 0.532 | **0.602** | **0.599** | **0.608** | **0.620** | **0.622** |

Table 3: Causal Direction Prediction Performance on Synthetic and Real Datasets. The performance metric is AUC (higher is better). We compare five variants of our proposed **CDCI** (Conditional Divergence based Causal Inference) approach namely CCS, CHD, CKL, CKM, and CTV with four popular baselines: ANM (Mooij et al., 2016), CDS (Fonollosa, 2019), IGCI (Daniušis et al., 2010), and RECI (Blöbaum et al., 2019). *Italics*: best performance of baselines, **bold**: better than the best baseline, red: best performance among all.

