# OpenReview forum: "Bivariate Causal Discovery via Conditional Divergence"
_cclear.cc/CLeaR/2022/Conference — CLeaR 2022 Poster_

### Official Review · Reviewer_hk9s · 2021-11-19

**Confidence:** 3
**Overall Score:** 7

**Main Review:**

Strenghts:
- The method is simple to understand, and applies to a wide range of bivariate causal discovery settings.
- The improve performance against state-of-the-art seems significant.

As mention in the conclusion, theoretical identifiability results and extension to multivariate causal discovery would be greatly appreciated:
- In terms of theory, it seems that your assumption in section 3.1 is not sufficient. Indeed, for additive linear Gaussian noise model, which is not identifiable, all X, Y, X|Y and Y|X are Gaussian, hence C_{X->Y} = 0 while there may exist some causal dependence in any direction. A deeper analysis of which settings give an actual asymmetry between cause and effect according to this criterion would be interesting.
- In terms of extension to the multivariate setting, it looks like the current binning approach for computing the divergences would highly suffer from the curse of dimensionality. However, this work builds an interesting basis for future work.

**Summary:**

This paper proposes a bivariate causal discovery method from purely observational data based on the assumption that the conditional distribution of the effect given the cause has the same shape for any value of the cause. It implicitly assumes that, on the other hand, conditioning the cause on the effect would not have this property. By exploiting this condition, they propose the conditional divergence based causal score, which measures how much the shape of the conditional distribution Y|X changes for various values of X, compared to that of X|Y. If X|Y changes more that Y|X, it is then concluded that X causes Y, and vice versa. The shape variation of Y|X is computed by estimating a divergence measures between the distributions of Y and Y|X after standardizing the two distributions, i.e., equating the first and second order moments. The estimation of this quantity is performed using a binning approach on the data. Various divergence measures are tested on synthetic and real world data, and show improved performance against state-of-the-art algorithms. The influence of the choice of divergence measure and other hyper-parameters, is shown to be small.

---

> ### Author Response · Authors · 2021-12-04
> **Reply to Reviewer hk9s**
>
> We greatly thank Reviewer hk9s for offering such thorough and constructive feedback, which we all agree. Regarding your concern about our assumption being insufficient to distinguish the causal direction in the case of linear non-Gaussian additive noise models, we will include it as part of a section discussing trivial identifiable and unidentifiable cases using our method. We also highly appreciate your insightful thoughts on how multivariate causal discovery can benefit from this study.

---

### Official Review · Reviewer_oj7Q · 2021-11-19

**Confidence:** 4
**Overall Score:** 7

**Main Review:**

The paper introduces a novel approach to discover causal effect directionality in the bivariate case. Overall, the paper is clearly written and the presented approach can be considered of high relevance for the audience of causal machine learning and related areas. However, there are some aspects (summarized below) that require additional attention prior to recommending acceptance of the paper in the Proceeding of Machine Learning Research.

p. 2 (end of introduction section):  For sake of completeness, I suggest including a class of causal discovery algorithms proposed by Peters and Bühlmann (2014) which identifies causal structures using an equal variance assumption (see also Chen et al., 2019 and Park and Kim, 2020).

pp. 3 & 4: The authors state that the functional causal model under consideration allows the presence of dependence between the cause (X) and the error of the causal model. Therefore, the functional causal model allows the presence of hidden confounders (Z). Here, it is currently unclear whether the presented algorithm is able to detect the true causal direction despite the presence of confounding or whether the algorithm is simply able to distinguish an unconfounded causal model (e.g., X → Y) from a confounder model (similar to previous approaches discussed, e.g., by Wiedermann and Li, 2018 and Maeda and Shimizu, 2020; see also Wiedermann & von Eye, 2016). Formal model definitions presented on p. 3 suggest the former. However, on p. 4, the authors state that only three classes are considered (X → Y, Y → X and a non-causal class X ← Z → Y) which suggests the latter. Further clarifications on the exact nature of the confounder model that can be present in CDCI are needed.

p. 4: The introductory section on the asymmetry between cause and effect seems incomplete for two reasons: First, in discussing the historical origins of cause-effect asymmetry modeling, the authors do not mention the work by Dodge and Rousson (2000, 2001) who were, to the best of my knowledge, among the first ones that suggested to determine causation from distributional features of observational data. Second, the section does not discuss copula-based approaches to determine cause-effect asymmetry (see, e.g., Kim & Kim, 2014; Sungur, 2005). In contrast to the presented algorithms, copula-based approaches may have the advantage that one does not have to select between causally competing models (such as X → Y or Y → X). Instead, dependence coefficients are estimated for both causal directions allowing to detect potential reciprocal relations. The present paper would benefit from incorporating these perspectives.

On a related note, the authors currently focus on the two (unidirectional) causal models, X → Y and Y → X, and a non-causal (confounder) model (X ← Z → Y) to explain the association between X and Y. In practice, however, at least two further explanations exist: The presence of a collider and the presence of a reciprocal relation. Can one expect that CDCI performs acceptably under these causal mechanisms as well?

p. 6, In the equation below Eq (1), what does the superscript (k) refer to here? Please specify.

pp. 6 & 7: The role of hyperparameters, in particular, the degree of discretization, should be discussed early on when introducing the divergence based causal score and the different variants of CDCI.

p. 10: Currently it is unclear why the authors selected ANM, CDS, IGCI, and RECI as competitors to the proposed CDCI approach. The competing methods differ in their underlying assumptions and in the utilized criteria to discern decisions concerning the causal direction of effect. Here, additional clarifications are needed. Also, the authors should discuss whether divergent assumptions of the competing methods may have unfavorably impacted their performance in the synthetic and real-world datasets.

Minor Issues:

p. 2, 1st paragraph, 2nd line from below: The authors mention that advances in causal machine learning enable one to detect causal structures even with “…unstructured data”. What do the authors mean with “unstructured”? Non-experimental/observational data? Please clarify.

p. 4: The authors state that the proposed method is able to classify a bivariate relation into three main categories and list four potential outcomes (X → Y, Y → X, non-causal relation, and independence). Please clarify.

p. 4, 2nd line from below: The authors mention that RECI is able to detect causal structures in “… the presence of a functional form”. It is unclear whether the authors refer to “knowledge” of the true functional form or the presence of a non-linear functional form. Please clarify.

p. 12 (1st line of Conclusion section): Please use “that a causal mechanism” instead of “that causal mechanism”.

References:

Chen, W., Drton, M., & Wang, Y. S. (2019). On causal discovery with an equal-variance assumption. Biometrika, 106(4), 973–980. https://doi.org/10.1093/biomet/asz049
Dodge, Y., & Rousson, V. (2000). Direction dependence in a regression line. Communications in Statistics-Theory and Methods, 29(9–10), 1957–1972. https://doi.org/10.1080/03610920008832589
Dodge, Y., & Rousson, V. (2001). On asymmetric properties of the correlation coefficient in the regression setting. The American Statistician, 55(1), 51–54. https://doi.org/10.1198/000313001300339932
Kim, D., & Kim, J.-M. (2014). Analysis of directional dependence using asymmetric copula-based regression models. Journal of Statistical Computation and Simulation, 84(9), 1990–2010. https://doi.org/10.1080/00949655.2013.779696
Maeda, T. N., & Shimizu, S. (2020). Causal discovery of linear non-Gaussian acyclic models in the presence of latent confounders. ArXiv: 2001.04197.
Park, G., & Kim, Y. (2020). Identifiability of Gaussian linear structural equation models with homogeneous and heterogeneous error variances. Journal of the Korean Statistical Society, 49(1), 276–292. https://doi.org/10.1007/s42952-019-00019-7
Peters, J., & Bühlmann, P. (2014). Identifiability of Gaussian structural equation models with equal error variances. Biometrika, 101(1), 219–228. https://doi.org/10.1093/biomet/ast043
Sungur, E. A. (2005). A note on directional dependence in regression setting. Communications in Statistics—Theory and Methods, 34(9–10), 1957–1965. https://doi.org/10.1080/03610920500201228
Wiedermann, W., & von Eye, A. (2016). Statistics and causality: Methods for applied empirical research. Hoboken, NJ: John Wiley & Sons.
Wiedermann, W., & Li, X. (2018). Direction dependence analysis: A framework to test the direction of effects in linear models with an implementation in SPSS. Behavior Research Methods, 50(4), 1581–1601. https://doi.org/10.3758/s13428-018-1031-x



**Summary:**

The main contribution of the paper is an introduction into a novel criterion, Conditional Divergence based Causal Inference (CDCI), to determine the causal direction of effects, i.e., whether an effect is transmitted from X to Y or vice versa, using (nonexperimental) observational data. The core idea of CDCI is that, under a causal mechanism of the form X → Y, the conditional distribution of the outcome (the effect) given the predictor (the cause), P(Y | X), will be independent of the marginal distribution of X, P(X). From this it follows that the shape of the conditional probability P(Y | X = x) can be expected to be invariant for different values x. Stability of P(Y | X = x) is quantified via divergence measures of the conditional distribution. In essence, the model with the smaller divergence of conditional distributions is selected. Overall, five variations of CDCI (with different distance measures) are proposed and their effectiveness is evaluated using synthetic and real-world data. Further, CDCI variants are contrasted with four state-of-the-art causal discovery methods. Results suggest that, in the majority of cases, CDCI outperformed the four selected state-of-the-art causal discovery methods.

---

> ### Author Response · Authors · 2021-12-04
> **Reply to Reviewer oj7Q (1/2)**
>
> We are greatly grateful for your time to review our paper and offer such detailed comments, as well as valuable suggestions. The shortcomings you pointed out are indeed relevant to this study and we will bridge those gaps in later versions, including approaches based on the equal variance assumption and a more thorough review. Below we give clarifications regarding your questions in your listed order:
>
> > Q1: pp. 3 & 4: The authors state that the functional causal model under consideration allows the presence of dependence between the cause (X) and the error of the causal model. Therefore, the functional causal model allows the presence of hidden confounders (Z). Here, it is currently unclear whether the presented algorithm is able to detect the true causal direction despite the presence of confounding or whether the algorithm is simply able to distinguish an unconfounded causal model (e.g., X → Y) from a confounder model (similar to previous approaches discussed, e.g., by Wiedermann and Li, 2018 and Maeda and Shimizu, 2020; see also Wiedermann & von Eye, 2016). Formal model definitions presented on p. 3 suggest the former. However, on p. 4, the authors state that only three classes are considered (X → Y, Y → X and a non-causal class X ← Z → Y) which suggests the latter. Further clarifications on the exact nature of the confounder model that can be present in CDCI are needed.
>
> - As for the nature of the confounder model involved in CDCI, we confirm that the algorithm aims to detect the true causal direction despite the presence of confounders. In Section 4, the “non-causal” class includes both cases X ← Z → Y and $X \perp Y$.
>
> > Q2: On a related note, the authors currently focus on the two (unidirectional) causal models, X → Y and Y → X, and a non-causal (confounder) model (X ← Z → Y) to explain the association between X and Y. In practice, however, at least two further explanations exist: The presence of a collider and the presence of a reciprocal relation. Can one expect that CDCI performs acceptably under these causal mechanisms as well?
>
> - About your concern on the effectiveness of our method in the presence of colliders, since we rely on the  i.i.d. assumption, the existence of a collider between X and Y should not affect the independence between them (if they are not confounded), which can still be detected with our method. We will consider mentioning this point in our revised version.
>
> - As for the reciprocal relation, indeed in this study we do not concern bi-directional causal relationships, which can be fundamentally different in assumptions and properties. We agree that this scenario surely presents in practice and is an interesting problem, thus we will highlight this as a future direction for research in the next version.
>
> > Q3: p. 6, In the equation below Eq (1), what does the superscript (k) refer to here? Please specify.
>
> - In the equation below Eq. (1), the superscript ($k$) indicates the number of the sample, i.e., $(x^{(k)},y^{(k)})$ refers to the $k$-th observation.
>
> > Q4: pp. 6 & 7: The role of hyperparameters, in particular, the degree of discretization, should be discussed early on when introducing the divergence based causal score and the different variants of CDCI.
>
> - We agree with your concern about this and will reorganize the manuscript according to your suggestion.
>
> > Q5: p. 10: Currently it is unclear why the authors selected ANM, CDS, IGCI, and RECI as competitors to the proposed CDCI approach. The competing methods differ in their underlying assumptions and in the utilized criteria to discern decisions concerning the causal direction of effect. Here, additional clarifications are needed. Also, the authors should discuss whether divergent assumptions of the competing methods may have unfavorably impacted their performance in the synthetic and real-world datasets.
>
> - On the choice of competitors to our approach, we choose these methods for two main reasons. Firstly, they are all recognized methods in the problem of predicting bivariate causal direction and have also been used as competitors in several closely similar studies (Goudet et al., 2018; Ton et al., 2021; Wu and Fukumizu, 2020). Secondly, they all have open source implementations (Kalainathan et al. 2020), which make experiments replicable and consistent.

---

> > ### Author Response · Authors · 2021-12-04
> > **Reply to Reviewer oj7Q (2/2)**
> >
> >
> > > Q6: p. 2, 1st paragraph, 2nd line from below: The authors mention that advances in causal machine learning enable one to detect causal structures even with “…unstructured data”. What do the authors mean with “unstructured”? Non-experimental/observational data? Please clarify.
> >
> > - By “unstructured data”, we refer to images, text, audio, etc., which are in contrary with tabular (structured) data.
> >
> > > Q7: p. 4: The authors state that the proposed method is able to classify a bivariate relation into three main categories and list four potential outcomes (X → Y, Y → X, non-causal relation, and independence). Please clarify.
> >
> > - Regarding your concern about the categories of causal relationships involved in this study, there are three classes, namely X causes Y, Y causes X (despite the presence of confounders), and “others” (this class includes both X ← Z → Y and $X \perp Y$ and we do not distinguish between them).
> >
> > > Q8: p. 4, 2nd line from below: The authors mention that RECI is able to detect causal structures in “… the presence of a functional form”. It is unclear whether the authors refer to “knowledge” of the true functional form or the presence of a non-linear functional form. Please clarify.
> >
> > - By “… the presence of a functional form”, we refer to the knowledge of a non-linear functional form.
> >
> > Thank you for pointing out these issues and the manuscript will be revised with all the clarifications above.
> >
> > **References**
> >
> > Olivier Goudet, Diviyan Kalainathan, Philippe Caillou, Isabelle Guyon, David Lopez-Paz, and Michele Sebag. Learning functional causal models with generative neural networks. In Explainable and Interpretable Models in Computer Vision and Machine Learning, pages 39–80. Springer, 2018.
> >
> > Jean-Francois Ton, Dino Sejdinovic, and Kenji Fukumizu. Meta learning for causal direction. In Proceedings of the AAAI Conference on Artificial Intelligence, pages 9897–9905, 2021.
> >
> > Pengzhou Wu and Kenji Fukumizu. Causal mosaic: Cause-effect inference via nonlinear ICA and ensemble method. In Proceedings of the International Conference on Artificial Intelligence and Statistics, pages 1157–1167, 2020.
> >
> > Diviyan Kalainathan, Olivier Goudet, and Ritik Dutta. Causal Discovery Toolbox: Uncovering causal relationships in Python. Journal of Machine Learning Research, pages 37–1, 2020.

---

### Official Review · Reviewer_DoYa · 2021-11-25

**Confidence:** 3
**Overall Score:** 9

**Main Review:**

Originality: The paper provides a new idea and method. Exploiting stability in shape was not exploited before to detect causal relationships, as far as I know (but Fonollosa 2019). The notion of conditional divergence is new.

Significance: The paper addresses an important and relevant problem for the CLeaR community. It is likely that it will also have an impact outside the CLeaR community.

The proposed approach is technically sound. However, it should be mentioned that some SCMs may not satisfy the assumption of stable causal mechanism. The possible violations of this assumption should be better highlighted.

Technical quality: The proposed approach is technically sound.

Clarity: The paper is clearly written and well-organised.

**Summary:**

The paper proposes a new method to identify causal direction between two variables exploiting the condition that the conditional distribution of effect given cause in invariant in shape.

---

> ### Author Response · Authors · 2021-12-04
> **Reply to Reviewer DoYa**
>
> We sincerely thank Reviewer DoYa for your time and insightful comments, as well as a supportive assessment for our work. We agree that possible violations of our assumption should be emphasized and we will consider presenting in the next version a wide range of examples for causal mechanism families that match and do not match our assumption.

---

### Decision · Program_Chairs · 2022-01-12

**Decision:**

Accept (Poster)

**Comment:**

The authors propose an observational bivariate causal discovery method when it is assumed that the conditional distribution of the effect given the cause has the same shape for any value of the cause using a conditional divergence-based causal score. All reviewers felt the paper is well written and the contribution is adequately novel. Additionally, found for the most part, the comparison against state-of-the-art is significant.